# The Indirect Effects of Fathers’ Parenting Style and Parent Emotion Regulation on the Relationship Between Father Self-Efficacy and Children’s Mental Health Difficulties

**DOI:** 10.3390/ijerph22010011

**Published:** 2024-12-25

**Authors:** Alicia Carbone, Carmela Pestell, Thom Nevill, Vincent Mancini

**Affiliations:** 1School of Psychological Science, The University of Western Australia, Crawley 6009, Australia; 23249169@student.uwa.edu.au (A.C.); carmela.pestell@uwa.edu.au (C.P.); 2The Kids Research Institute Australia, Nedlands 6009, Australia; thom.nevill@thekids.org.au

**Keywords:** father self-efficacy, parenting styles, child mental health, parent emotion regulation

## Abstract

Improving parental self-efficacy has been linked with reductions in child mental health difficulties; however, underlying mechanisms remain unclear, especially for fathers. This study investigated whether father self-efficacy influences child mental health difficulties indirectly through parenting style and parent-facilitated regulation of children’s negative emotions. A community sample of American fathers (*N* = 350, M = 39.45 years old) completed self-reports on father self-efficacy, parenting styles, parent-facilitated emotion regulation, and their children’s mental health difficulties (aged 4–12). Path analysis was used to test a cross-sectional, parallel–sequential indirect effect model. Father self-efficacy had a significant indirect effect on child mental health difficulties via three significant pathways of permissive parenting, authoritative parenting–acceptance of child’s negative emotions, and authoritarian parenting–avoidance of child’s negative emotions. Our model explained a moderate amount of variance in child mental health difficulties. The findings support promoting father self-efficacy through parenting interventions and highlight parenting beliefs as important for clinicians providing child mental health care.

## 1. Introduction

Early research investigating the influence of parenting on children’s development traditionally focused on mothers, potentially overlooking the unique contributions of fathers [1,2]. Even today, paid parental leave policies see fathers as ‘secondary caregivers’, as approximately half of the US Fortune 500 companies offer women twice as much leave as men [3]. Such policies reinforce outdated gendering of parent roles, where mothers are responsible for childcare and fathers are responsible for financially providing [3,4].

However, the sociocultural landscape of parenting is changing due to increases in dual-earner households (two-parent families), where mothers and fathers both provide for the family economically and are engaged in active childcare [4]. This is reflected in Australian longitudinal data revealing a trend towards more engaged parenting as fathers’ active childcare time has increased between 1996 and 2006 and mothers have remained the same, despite increased workloads [5]. Contemporary parenting theories situate maternal and paternal roles as ‘complementary’ in overall child-rearing [6,7] (p. 349). However, paternal influences on child development remains under-developed due to a remaining over-reliance on the ‘maternal template’ in the parenting literature [4] (p. 348).

This has led to a considerable underrepresentation of fathers in parenting studies and reduced effectiveness for improving fathers’ parenting behaviours in parenting interventions (e.g., Triple P Parenting Program) as compared to mothers [8]. This suggests a potential mismatch between parenting programme content and fathers’ intervention needs. Furthermore, fathering measures are also commonly derived from maternal measures, potentially missing salient components in the father–child dyad (e.g., positive engagement activities with the child) [4,9]. Thus, there is a need for father-focused research with validated fathering measures to understand the paternal influences on child development.

Fathers play a critical role in uniquely influencing child wellbeing [10]. Poor mental health conditions in childhood cause a multitude of problems for children, families, and communities [11]. Approximately 13.4% of the world’s children and adolescents are thought to have a mental health condition [12], and adults with mental health conditions are usually first diagnosed as young as 14 years old [13,14]. Policies such as the Australian National Children’s Mental Health and Wellbeing Strategy prioritise supporting families, particularly parents, to promote positive child mental health outcomes [15]. There is emerging research into healthy parenting behaviours that fathers can utilise to assist in reducing this risk of child mental health difficulties [16]. Meta-analytic research indicates that mid-late childhood is an optimal period for improving parenting behaviours, as modifying behaviours of older children (adolescents) becomes more difficult [17]. Research identifying protective parenting behaviours in fathers of the mid-late childhood age group is important to optimise caregiving environments and positive child mental health outcomes.

### 1.1. Fathering Theories

In theories on fathering, fathers are theorised to be part of a unique ‘activation’ relationship with their children, according to Paquette [18] (p. 202). Fathers are thought to encourage healthy risk taking in children, equip children to ‘stand up for themselves’ and spark a child’s curiosity to new opportunities [18] (p. 212). This ‘activation’ relationship is thought to be developed through physical play with children, also known as ‘rough-and-tumble play’ [18]. A recent fathering study by Majdandzic, de Vente, Colonnesi and Bogels [19] builds on this and suggests fathers who promote challenging parenting behaviours (i.e., lean into the father–child ‘activation relationship’) have children who are less likely to develop mental health symptoms such as anxiety in early childhood. This demonstrates the unique influence of actively involved fathers on child mental health outcomes.

The Paternal Involvement theory by Pleck [20] proposes involved fathers participate in their child’s life through a combination of positive engagement activities, positive parenting behaviours (warmth, responsiveness, and control), indirect care (e.g., making childcare arrangements), and consistently monitoring their child’s needs through ‘process responsibility’ [20] (p. 88). These parenting behaviours are theorised to foster healthy child developmental outcomes [20]. This theory provides opportunities to test direct and indirect influences from fathers on child development, arguing that the optimisation of father involvement exists in many forms [20].

While Pleck [20] provides valuable insights, it lacks a family–systems context to situate the father–child dyad and does not address how fathers’ personal parental characteristics help shape child development. These external factors become important and can either promote or hinder the familial caregiving environment and health of the child [21]. Thus, Cabrera, Fitzgerald, Bradley, and Roggman [6] advance Pleck’s theory and provide a dynamic model of fathering situated in a wider context called The Father-Child Relationships Expanded Model, as seen in Figure 1. This model provides pathways of paternal influence on child mental health development, such as a father’s personal characteristics (e.g., parenting beliefs) influencing parenting behaviours and subsequently influencing child development, shown in the highlighted white boxes in Figure 1.

These two theories align through understanding how fathers provide indirect influences on child development through positive parenting behaviours (parental warmth, control, and responsiveness), and demonstrate multi-faceted approaches to promoting quality fathering. However, the Cabrera, Fitzgerald, Bradley, and Roggman [6] model advance this by providing the possibility to test relationships within a wider family context. By utilising these theories, we can investigate how fathers’ parenting can indirectly affect and potentially protect children from developing mental health difficulties.

### 1.2. Parental Self-Efficacy

Unlike more rigid mental health risk factors such as socioeconomic status [22], parenting beliefs and parenting behaviours can be modified through interventions (e.g., by mental health professionals) and be effective in improving child mental health outcomes [8,23,24]. Parental self-efficacy (PSE) is the belief a parent has in their ability to perform demanding child-rearing tasks and positively influence their child’s health and future success [25,26]. PSE originates from Bandura’s social cognitive theory [27] and is perceived by fathers as being influenced by parent role-modelling, past parenting experiences, personal beliefs, and positive reinforcement from the child [28].

Research shows father self-efficacy (PSE for fathers) rather than co-parenting quality, statistically predicts father involvement in a bi-directional relationship [29]. Targeting father self-efficacy could promote positive paternal involvement in children’s developmental lives [29]. Further longitudinal research supports PSE as a predictor for influencing parent behaviours and adolescent externalising behaviours, rather than an adolescent-driven or parent-behaviour-driven model [30]. This is consistent with findings that improving PSE produces ‘cascading’ beneficial effects on parenting behaviours, showing medium to high effect sizes [31] (p. 156). These findings suggest father self-efficacy could be harnessed as a powerful cognitive tool for influencing parent behaviours and child mental health symptoms. With validated father self-efficacy measures becoming available [9], there is a possibility to test direct and indirect influences of father self-efficacy on parenting behaviours and child outcomes.

### 1.3. Parenting Behaviours

Parent behaviours shape children’s interactions with the environment, and the childrearing environment itself [32]. Positive healthy parent behaviours reduce children’s risk of poor mental health [33], and aversive parenting behaviours increase this risk through manifestations such as anxiety and depression [34,35].

Interventions directly targeting PSE have shown effective reductions in child internalising and externalising symptoms through indirectly influencing parent behaviours. For example, the Confident Parents Belgium programme guided parents of young children (aged 3–6) with externalising problems through 8-week intervention sessions and saw parents with greater PSE reporting significant reductions in their child’s externalising problems, with moderate–large effect sizes (*d* = 0.62) [23]. Similarly, a meta-analysis of 30 years of the Triple P Parenting Program demonstrated improvements in PSE led to improvements in fathers’ parenting practices (*d* = 0.35) and subsequent child socio-emotional wellbeing outcomes (*d* = 0.38), with small to medium-sized effects [36]. Therefore, these intervention studies are promising in demonstrating the modifiable and influential nature of PSE on child mental health. However, the mechanisms by which fathers uniquely influence child wellbeing remains under-researched.

### 1.4. Parenting Styles

The well-researched and influential theory on parenting behaviour by Baumrind [37] encompasses three typologies of parenting styles that include authoritative parenting (high warmth, high control), authoritarian parenting (low warmth, high control), and permissive parenting (high warmth, low control). In addition to these three parenting styles, a fourth parenting style is known as negligent parenting or uninvolved parenting [38]. Unlike the three parenting styles by Baumrind [37] which continue to dominate the literature, negligent parents are difficult to engage in parenting research and will not be the focus of this study due to this challenge. The authoritative parenting style has been found to be significantly associated with lower child mental health symptom severity than both authoritarian and permissive parenting styles [39]. Meanwhile, authoritarian or angry parenting has been associated with adolescent internalising problems (e.g., anxiety and depression) [22,40], and permissive parenting has been associated with child internalising and externalising problems, to a lesser degree than authoritarian parenting [41]. Perhaps a reason for this could be that authoritative parents grant autonomy and use democratic discipline only when required, while authoritarian and permissive parents restrict the child’s autonomy through forceful discipline and overprotection, respectively [37,42]. Understanding how to promote authoritative parenting in fathers and intervene in poorer parenting styles is crucial in order to provide beneficial mental health outcomes for children.

Research has posited parenting styles as an indirect effect variable between PSE and child mental health outcomes in a Korean study of fathers with school-age children [43]. A father’s PSE had a significant indirect effect on their child’s mental health (both internalising and externalising problems) through the variable of parental behaviour [43]. Increased father PSE increased warm parenting behaviours and decreased child internalising and externalising symptoms [43]. Although not an indirect effect on child mental health, increased PSE in fathers was also significantly associated with positive controlling parenting behaviours [43]. Thus, intervening in PSE could provide a possible protective mechanism for child mental health difficulties through the promotion of authoritative parenting styles. While this study provides robust findings, we can advance our understanding of parent behaviour influences by incorporating parenting styles with another salient parenting behaviour.

Parenting styles are also thought to be associated with child emotion regulation [44]. Research suggests that positive parenting behaviours (e.g., authoritative parenting) is significantly associated with better emotion regulatory skills in parents and children, while harsher parenting (e.g., authoritarian parenting) is associated with poorer emotion regulatory skills in parents and children [45]. Although past studies have demonstrated small effect sizes for a direct effect, potential indirect pathways may demonstrate emotion regulation as another salient factor in how parents—particularly fathers—may shape children’s mental health [44,45,46].

### 1.5. Parent-Facilitated Emotion Regulation

Emotion regulation is defined as having a sense of control over one’s emotions through monitoring, investigating, and changing one’s emotional reactions in line with one’s goals [45]. It is known that children who learn poorer emotion regulation strategies are at greater risk of developing anxiety, depression, and problem behaviours across childhood [44,47,48].

Fathers have been found to be uniquely influential in children’s expressions of emotions [49]. Research has demonstrated that the security of the attachment relationship in the father–child dyad significantly predicts children’s emotion regulation [50]. This secure attachment promotes more adaptive child emotion regulation and the ability to emotionally cope with stress from toddlerhood up to adolescence [51].

Children develop their capacity to regulate emotion through their parent’s reactions to their own (child’s) negative emotions, known as parent-facilitated emotion regulation. According to Pereira et al. [52], parents can endorse positive reactions to children’s negative emotions through (1) orientation towards the child’s emotions (e.g., encouraging their child’s emotional expression and problem solving) and (2) the acceptance of their own and their child’s emotions (e.g., some toleration of negative emotionality). By contrast, parents can endorse negative reactions to children’s negative emotions through (3) avoidance of the child’s emotions (e.g., minimising and distracting the child from negative emotions) and (4) a lack of emotional control (of their own emotions in front of the child) [52]. Parent orientation to emotions has been linked with promoting positive emotion regulation strategies in children and fewer internalising symptoms [53]. However, there is limited research on the other parent emotion regulation strategies in fathers, with inconclusive evidence of the outcomes of the avoidance of children’s emotions [52]. Further investigation is needed to understand the influences of the different parent-facilitated emotion regulation strategies proposed by Pereira, Barros, Roberto, and Marques [52] on fathers’ parenting behaviours and child mental health outcomes.

Like parenting styles, fathers’ emotion regulation is malleable to change through emotion-coaching interventions designed for fathers such as the Dad’s Tuning in to Kids programme [54]. This intervention has demonstrated medium-sized effects on reducing fathers’ negative parent emotion regulation strategies (e.g., minimising and critiquing) and increased positive strategies and responses (e.g., orientation) to children’s negative emotions, resulting in improved father-reported child socio-emotional functioning (small effect size) [54]. This intervention study has also highlighted the importance of fathers’ ‘modelling’ positive emotion regulation strategies, yet it is limited by the potential expectancy bias of fathers’ self-reports, warranting further research in a non-intervention context to clarify these links [46,54] (p. 30).

Overall, this research suggests the promotion of PSE in fathers may indirectly lead to more positive parenting and adaptive regulation of children’s emotions, in turn reducing the risk for poor mental health. However, no empirical evidence on this exists, particularly within a fathering cohort.

### 1.6. Conceptual Model

This study’s aim will be to determine if the relationship between father self-efficacy and child mental health difficulties is associated through the indirect process of parenting style and parent emotion regulation in a community sample of men. To examine this process, we put forward the following conceptual model hypotheses in Figure 2.

The model above serves as a conceptual guide for testing our four hypotheses below. Due to the structure of the parallel–sequential model, the hypotheses can be satisfied through various significant pathways, e.g., paths *a* → *c* in Hypothesis 1 denotes paths *a*_1_*c*_1_, *a*_2_*c*_2_, and path *a*_3_*c*_3_. The four hypotheses are displayed as follows:

**Hypothesis 1** **(H_1_):**
*Father self-efficacy has a significant total effect on children’s mental health difficulties (path f).*


**Hypothesis 2** **(H_2_):**
*Father self-efficacy has a significant indirect effect on children’s mental health difficulties through parenting style (paths *
*a*
*→*
*c*
*).*


**Hypothesis 3** **(H_3_):**
*Father self-efficacy has a significant indirect effect on children’s mental health difficulties through parent-facilitated emotion regulation (*
*paths b*
*→*
*e*
*).*


**Hypothesis 4** **(H_4_):**
*Father self-efficacy has a significant indirect effect on children’s mental health difficulties through the sequential variables of parenting style and parent-facilitated emotion regulation (paths *
*a*
*→*
*d*
*→*
*e*
*).*


## 2. Materials and Methods

### 2.1. Research Design

This current study utilised a quantitative cross-sectional design with a mixed parallel–sequential model. Father self-efficacy served as the predictor. Parenting styles (authoritative, authoritarian, and permissive) served as the first indirect effect variable in the sequence. Parent-facilitated emotion regulation (orientation to child’s emotions, avoidance of the child’s emotions, emotional lack of control, and acceptance of the child’s and parents’ emotions) served as the second indirect effect variable in the sequence. Children’s mental health difficulties served as the dependent variable. Although cross-sectional, support for the hypothesised sequential processes is informed by past research described in the prior introduction section. A priori analysis conducted revealed a required sample size of 148 participants for calculating moderate-sized effects for indirect effect variables with a power of 0.80, using 95% bias-corrected bootstrap confident intervals to identify statistical significance [55].

### 2.2. Participants

This study recruited 350 fathers aged 22 to 80 years old (*M* = 39.45 years, *SD* = 8.28) from the United States. The participants were recruited using the Prolific platform, which allows researchers to advertise online studies that eligible users may complete for monetary gain. The eligibility criteria for participating were being a father, stepfather, or adopted father; having English language competency; and living in the United States and having at least one child aged between 4 and 12 years old. A majority of the fathers were married (88.3%), part-time workers (71.1%), and had completed university studies (92.9%). The fathers answered questions in relation to one of their children aged between 4 and 12 years old. These children were equally male (49.7%) and female (50.3%) and were an average age of 7.5 years old (*SD* = 1.99). A comprehensive account of participant demographics is included (see Appendix A). Ethics for this current study was approved by the [name withheld for blind review] Human Research Ethics Committee (HREC).

### 2.3. Measures

#### 2.3.1. Demographic Questionnaire

Participants completed a questionnaire regarding their own demographic information including their birth country, age, relationship status, employment level, and education level. Participants also completed demographic questions related to their children, including the number of children they had and the children’s ages and sexes.

#### 2.3.2. Father Self-Efficacy Scale (FSES)

This study utilised the Father Self-Efficacy Scale (FSES) created by Sevigny, Loutzenhiser, and McAuslan [9], a self-report scale that assesses fathers’ self-efficacy beliefs about their parenting. Participants were presented with 22 items spanning across three subscales of father self-efficacy: (1) positive engagement, (2) direct care, and (3) financial responsibility. An example item from the positive engagement subscale is ‘I can always think of fun things to do with my child’. Participants responded to each item on a 9-point Likert scale, anchored from 1 (completely disagree) to 9 (completely agree). The item scores were averaged together to create three subscale scores; these were then averaged together to create the total FSES score. This FSES total score ranged between 22 and 198, with higher scores indicating greater self-efficacy. Previous research has found the FSES to have strong convergent validity with general self-efficacy and general parental self-efficacy, as well as good test–re-test reliability (*r* = 0.81) and internal reliability (α = 0.88) [9]. This demonstrates the FSES to be a suitable tool for measuring father self-efficacy in this current study.

#### 2.3.3. Parenting Styles and Dimensions Questionnaire—Short Form (PSDQ-SF)

This current study applied the short version of the PSDQ, the PSDQ-SF formulated by Robinson et al. [56]. The PSDQ-SF is a parental self-report questionnaire evaluating parenting behaviours categorised by three distinct parenting styles (authoritative, authoritarian, and permissive) developed by Baumrind [37]. The PSDQ-SF is a 32-item questionnaire spanning three parenting style subscales of authoritative parenting (15 items), authoritarian parenting (12 items), and permissive parenting (5 items). Participants respond on a 5-point Likert scale anchored from 1 (never) to 5 (always) on all items. Examples of authoritative, authoritarian, and permissive items are ‘I am responsive to our child’s feelings or needs’, ‘I yell or shout when our child misbehaves’, and ‘I give into our child when the child causes a commotion about something’, respectively.

Each parenting style subscale is averaged to produce a score ranging from 1 to 5, and higher scores indicate a greater adherence to the parenting style. As there is limited psychometric research of the PSDQ-SF in Western cultures, a translated version was conducted on mothers of children aged 3–18 in Brazil and was found to have suitable content validity and good internal reliability for the authoritative (ω = 0.86) and authoritarian parenting style subscales (ω = 0.84) [57]. Lower internal reliability for the permissive parenting style (ω = 0.64) subscale was found, potentially due to this scale only measuring 5 items [57]. Thus, the PSDQ-SF has been demonstrated to display good validity and reliability and is suitable for measuring parenting styles in this current study.

#### 2.3.4. Parent Emotion Regulation Scale (PERS)

The PERS, created by Pereira, Barros, Roberto, and Marques [52], was administered in this current study. The PERS is a parent self-report scale that evaluates a parent’s own ability to regulate their children’s negative emotions most of the time, designed initially for mothers of children aged 3 to 15 years old. The PERS is made up of 20 items divided into 4 sub-scales consisting of (1) orientation to child’s emotions (OCE), (2) avoidance of the child’s emotions (ACE), (3) emotional lack of control (ELC), and (4) acceptance of the child’s and parents’ emotions (ACPE). Participants respond to the PERS items on a 5-point Likert scale anchored from 0 (strongly disagree) to 4 (strongly agree). An example item for each of the four subscales are as follows: (OCE) ‘I am attentive to my child’s emotions and try and understand them’, (ACE) ‘If I could, I would eliminate all my child’s negative emotions’, (ELC) ‘I do cry or reveal myself as very sad and worried in front of my child’, and (ACPE) ‘When my child is nervous, I don’t act immediately and give him time to calm down or solve the situation’. The items in each subscale are averaged together to create subscale means. Previous research has found the PERS subscales to have suitable construct validity and acceptable internal reliabilities for OCE (α = 0.79) and ACE (α = 0.73), and a slightly lower internal consistency for ELC (α = 0.69) and ACPE (α = 0.62), likely due to the ACPE subscale being made up of only 4 items [52]. Thus, the PERS has demonstrated good reliability and validity and is suitable for measuring parent-facilitated emotion regulation of their children.

#### 2.3.5. Paediatric Symptom Checklist-17 (PSC-17)

This study administered the short version of the Paediatric Symptom Checklist, the PSC-17 [58]. The PSC-17 is a parental self-report scale used as a screening tool in identifying psycho-social problems in children aged 5–17 years old. Participants are prompted to respond in a way that best describes their child for each statement of the 17-item scale. Responses are measured on a 3-point Likert scale ranging from 0 (never) to 2 (often). An example item the participant responds to about their child is ‘blame(s) others for his or her troubles’. A total overall PSC-17 score is calculated by summing the 17 items together. If the PSC-17 total score is ≥15, a ‘positive score’ is interpreted and indicates further need for psychosocial evaluation of the child by a specialist. Previous research indicates the PSC-17 to have high internal reliability α = 0.89, demonstrated in a large robust study containing paediatric outpatients (*n* = 80,000) [59]. The PSC-17 scale has good convergent and divergent validity and is seen as a psychometrically sound tool for assessing psycho-social functioning in children [60].

### 2.4. Procedure

Participants were recruited through the Prolific (www.prolific.com: accessed on 28 June 2024) platform, which has been shown to obtain higher-quality research data than sampling platforms such as MTurk and SONA [61]. Eligible participants completed the survey on the Qualtrics (www.Qualtrics.com: accessed on 28 June 2024) platform. Before testing, participants read an information sheet, provided written informed consent, and were reminded to accurately respond to attention checks throughout the survey for the responses to be valid. Participants were then asked demographic questions and completed a randomised battery of 19 tests related to fathering and child development; however, only the four measures of PSDQ-SF, FSES, PERS, and PSC-17 were used in this current study, as the remaining tests were part of a larger study conducted by [name removed for blinding]. The survey took approximately 41 min to complete. After survey submission, participants were debriefed and compensated AUD ~$8. Participants completed the survey on a chosen device from any location and were able to withdraw at any time.

### 2.5. Data Analyses

This current study utilised the Lavaan Package in R and version 29 of SPSS to analyse all data [62]. SPSS was used to obtain descriptive statistics, correlations, internal reliabilities, and assumptions. This current study specified a two-step parallel–sequential indirect effect model assessing the relationship between father self-efficacy and child mental health via the indirect pathways of parenting style and parents’ emotion regulation of their children. Research suggests parenting behaviours can vary based on a child’s sex [63] and child’s age [64], and PSE can vary based on the number of children a parent has [65]. Thus, these variables (child’s sex and age, number of children) were controlled for in this current study.

The model proposed was tested using the Lavaan package in R to calculate the direct and indirect associations between the variables of interest [66]. Utilising a sequential model offers opportunities to test associations between two indirect effect variables occurring in a sequence that cannot be achieved through parallel models alone. Parenting styles (PSDQ-SF scores) of authoritative, authoritarian, and permissive parenting were included in step one. Parent emotion regulation (PERS scores) dimensions of orientation, acceptance, emotional lack of control and avoidance of emotions were included in step two of the model. Bias-corrected boot-strap confidence intervals (5000) were used to test the indirect effects of father self-efficacy on child mental health. Since the parallel–sequential model cannot distinguish causal influences, we did not specify the ‘indirect effect variables’ as ‘mediators’, as doing so could have biased the results [67] (p. 465).

## 3. Results

### 3.1. Data Cleaning and Assumption Checks

The original sample who accessed the survey consisted of 390 participants. Of these people, 25 participants failed to complete the entire survey, 4 participants did not satisfy the eligibility criteria, and 11 participants failed to meet the attention checks scattered throughout the study. Thus, the final dataset retained for analyses consisted of 350 participants.

Prior to interpreting results, four assumption checks for the parallel–sequential mediation analyses were conducted. These assumptions underpin regression-based analyses, including for analyses testing indirect effects between continuous variables [68]. First, inspection of skewness, kurtosis, and histograms for each primary variable approximated a normal distribution. This was indicated by the data being below the cut-off values for skewness (<2.00) and for kurtosis (<7.00), as recommended from guidelines by Kim [69]. Second, the absence of outlier’s assumption was checked for the primary variables, and appropriate values were observed for Cook’s distance (<1.00), indicating no cases with high leverage and no Mahalanobis Distance exceeding the critical value, satisfying this assumption. Third, the multicollinearity assumption was checked for all variables in the model. Observations of tolerance values indicated low multi-collinearity as values ranged between 0.48 and 0.98, well above the 0.20 threshold, and observations of the variance inflation factor (VIF) indicated that the variance was not inflated due to multi-collinearity, as values ranged between 1.02 and 2.10, well below the commonly used cut-off of 5.00. This suggests the multicollinearity assumption was satisfied. Last, the assumption of normality, linearity, and homoscedasticity of residuals was tested for in the variables in the model. Visual inspection of the histogram, plot of regression standardised residuals, and scatterplot of residuals indicated the residuals approximated normal distribution, were linear and mostly homoscedastic, and dispersed, suggesting that this assumption was satisfied. Given that assumptions were satisfied, we can proceed with interpreting the planned analysis.

### 3.2. Descriptive Statistics

The descriptive statistics as well as the skewness and kurtosis values described earlier are displayed in Table 1 for the primary and control variables. All measures in this current study demonstrated suitable internal reliability, yet lower internal reliability was found for the permissive parenting style, emotional lack of control, and acceptance of emotions variables. This was to be expected and remains consistent with previous studies, likely due to the lower number of items in these subscales [52,57].

### 3.3. Correlations

The unadjusted correlations are displayed in Table 2 for the primary and control variables. 

Most of the primary variables included in the model indicated significant moderate–strong correlations, apart from the PERS variables of avoidance of emotions and acceptance of emotions. Greater father self-efficacy was positively associated with authoritative parenting, orientation to emotions, and acceptance of emotions, and negatively associated with emotional lack of control, permissive and authoritarian parenting, and child mental health difficulties. The correlations concerning the PERS measure were surprising, as the acceptance of emotions and avoidance of emotions variables were not statistically significantly correlated with child mental health difficulties.

### 3.4. Predicting Child Mental Health Difficulties

All the variables in the parallel–sequential model illustrated in Figure 3, including the control variables, accounted for a significant 18.5% of the variance in the outcome variable of child mental health difficulties (*F* (9, 340) = 110.90, *p* < 0.001). This is indicative of a medium-sized effect according to guidelines by Cohen [70]. The model variances for all primary variables are displayed in Figure 3.

#### 3.4.1. Total, Direct, and Indirect Effects

The total effect of father self-efficacy on child mental health difficulties (e.g., the sum of the direct and all indirect pathways) was statistically significant (β = −0.09, *p* < 0.001, 95% CI [−0.12, −0.07]), in support of *H*_1_. The direct effect (i.e., effect of father self-efficacy on child mental health difficulties after accounting for all other variables) was reduced but remained statistically significant (β = −0.06, *p* = 0.004, 95% CI [−0.10, −0.02]), indicative of a partial indirect effect. This means an increase in father self-efficacy was associated with a decrease in child mental health difficulties, partly through associations with the intermediary variables.

The total indirect effect (i.e., sum of all possible indirect effects) was statistically significant (β = −0.04, *p* = 0.017, 95% CI [−0.06, −0.01]). Upon inspection of all indirect effects, there were three statistically significant partial indirect effects made up of the following pathways: the permissive parenting pathway, authoritative parenting–acceptance of emotions pathway, and authoritarian parenting–avoidance of emotions pathway, presented in Figure 3. These three pathways made up 37% of the variance in the total indirect effect; however, given that these pathways constituted both positive and negative pathways, this is an approximate measure of magnitude and needs to be interpreted with caution. This is due to the magnitude of the indirect effect likely being attenuated by combining positive and negative values.

#### 3.4.2. Indirect Effect Pathways

The Permissive Parenting Pathway: Most of the total indirect effect was accounted for by the singular variable of permissive parenting; this was a statistically significant indirect effect (β = −0.020, *p* = 0.01, 95% CI [−0.03, −0.01]). Lower father self-efficacy was associated with higher levels of the permissive parenting style, which in turn was associated with higher levels of child mental health difficulties. Given that only one of the three parenting styles acted as a significant indirect effect variable, H_2_ can be partially supported.

The Authoritative Parenting–Acceptance of Emotions Pathway: The second significant indirect pathway from father self-efficacy to child mental health difficulties was through the sequential indirect effect variables of authoritative parenting and acceptance of emotions (β = 0.004, *p* = 0.05, 95% CI [0.001, 0.008]), partially supporting H_4_. Higher father self-efficacy was associated with higher levels of the authoritative parenting style, which in turn was associated with greater acceptance of their child’s emotions and increased child mental health difficulties. However, this was surprising, as it was expected that greater acceptance of emotions as a positive parent emotion regulation strategy would be associated with a decrease in child mental health difficulties.

Further analysis revealed a potential suppression effect. Initially including authoritative parenting as an individual indirect effect variable resulted in a statistically significant direct effect (β = −0.086, *p* < 0.001, 95% CI [−0.12, −0.05]) and a non-significant indirect effect. However, including both sequential indirect effect variables of authoritative parenting and acceptance of emotions grew the direct effect in magnitude (β = −0.087, *p* < 0.001, 95% CI [−0.12, −0.06]), and the indirect effect became significant. According to MacKinnon, Krull, and Lockwood [71], suppression can be seen when the inclusion of a third variable strengthens the relationship between an independent variable and a dependent variable. In the given study, this can be seen with the addition of the second mediator (acceptance of emotions), strengthening the relationship between the independent variable (father self-efficacy) and dependent variable (child mental health difficulties).

The Authoritarian Parenting–Avoidance of Emotions Pathway: The third significant indirect pathway from father self-efficacy to child mental health difficulties was through the sequential variables of authoritarian parenting and avoidance of emotions (β = 0.003, *p* = 0.05, 95% CI [0.0, 0.01]), partially supporting H_4_. Lower father self-efficacy was associated with higher levels of authoritarian parenting style, increased avoidance to emotions, and decreased child mental health difficulties.

Further analysis did not reveal a potential suppression effect. Initially including the authoritarian parenting as an individual indirect effect variable resulted in a statistically significant direct effect (β = −0.083, *p* < 0.01, 95% CI [−0.01, −0.06]) and a non-significant indirect effect. However, including both authoritarian parenting and avoidance of emotions as sequential indirect effect variables led to a smaller-in-magnitude direct effect that remained statistically significant (β = −0.81, *p* < 0.01, 95% CI [−0.12, −0.31]), and the indirect effect pathway became significant.

There were no significant indirect effects from father self-efficacy to child mental health difficulties for each of the parent emotion regulation strategies, as seen for orientation to emotion (β = −0.005, *p* = 0.33, 95% CI [−0.02, 0.01]), acceptance of child’s and parents emotions (β = 0.001, *p* = 0.67, 95% CI [−0.003, 0.0]), avoidance of child’s emotions (β = −0.002, *p* = 0.40, 95% CI [−0.01, 0.002]), and emotional lack of control (β = −0.004, *p* = 0.52, 95% CI [−0.02, 0.01]). Thus, H_3_ was not supported.

## 4. Discussion

This current study aimed to determine if the relationship between father self-efficacy and child mental health difficulties was associated, through the indirect process of parenting style and parent emotion regulation in a community sample of men. Consistent with this aim, our study raised four hypotheses that were tested using path analysis in a cross-sectional parallel–sequential model. The study controlled for the child’s sex and age and the number of children the father had. The findings will be discussed, along with theoretical and practical implications, limitations, and future directions for research.

Our first hypothesis (H_1_) that father self-efficacy would have a significant total effect (path *f*) on child mental health difficulties was supported and was partly made up of a significant direct (path *f’*) and indirect effect. The parallel–sequential model accounted for a medium-sized effect, a significant 18.5% of the variance in child mental health difficulties. Overall, we found an inverse relationship between father self-efficacy and child mental health difficulties where increased father self-efficacy was associated with a decrease in child mental health difficulties, partly through associations with parenting style and parent emotion regulation. This aligns with prior research demonstrating a moderate-sized effect between PSE and child behavioural problems [72]. However, the inverse relationship between PSE and child mental health difficulties has typically been found in mothers (excluding intervention research) and not for fathers [30,73]. This could be due to these samples being made up of mostly mothers than fathers and utilising PSE measures originally designed for mothers while missing salient PSE components of fathers. Thus, these findings contribute to the growing parenting and fathering literature.

Our second hypothesis (H_2_) that father self-efficacy would have a significant indirect effect on child mental health difficulties through parenting style (paths *a* → *c*) was partially supported with one significant indirect pathway through the permissive parenting style. Lower father self-efficacy was associated with increased permissive parenting and increased child mental health difficulties. This supports prior research of an inverse significant relationship between PSE and permissive parenting [26], as well as a positive significant relationship between permissive parenting and child internalising symptoms (e.g., generalised anxiety) [74] and externalising symptoms (e.g., ADHD) [41,75]. Our study uniquely connected these variables into one significant permissive parenting pathway in fathers. Overall, this finding suggests that less confident fathers may utilise permissive parenting (high warmth, low control) and over-protective behaviours to make their child happy short-term, yet the lack of discipline and child autonomy may lead to poorer child mental health outcomes [26,37,42].

Our third hypothesis (H_3_) that father self-efficacy would significantly indirectly influence child mental health difficulties through each of the four parent emotion regulation strategies (paths b → e) was not supported. However, the unadjusted correlations revealed that father orientation to emotions was significantly inversely associated with child mental health difficulties (r = −0.27, *p* < 0.001), consistent with prior research on father orientation to emotions and child internalising symptoms (r = −0.26, *p* < 0.001) [53]. Additionally, father emotional lack of control had a positive significant association with child mental health difficulties (r = 0.24, *p* < 0.001). Despite both father orientation of emotions and emotional lack of control not being a part of a significant indirect effect pathway from father self-efficacy to child mental health difficulties, the many significant correlations between these variables and others (e.g., parenting styles) within this study warrant further investigation. Furthermore, research posits an additional salient indirect effect variable of child emotion regulation between parent emotion regulation and child internalising symptoms [76,77]. Fathering research could also incorporate child emotion regulation into future models to better understand the father–child dyad.

Our fourth hypothesis (H_4_) that father self-efficacy would have a significant indirect effect on child mental health difficulties sequentially through parenting style and parent regulation of children’s negative emotions (paths *a* → *d* → *e*) was partially supported. This was through two significant pathways: the authoritative parenting–acceptance of emotions pathway and the authoritarian parenting–avoidance of emotions pathway. Given that parental acceptance and avoidance of children’s emotions were not significantly correlated with child mental health difficulties prior to path analysis, the combined serial influence of parenting style and parent emotion regulation is important.

The authoritative parenting–acceptance of emotions pathway was such that higher levels of father self-efficacy were associated with increased authoritative parenting, which in turn was associated with greater acceptance of their child’s negative emotions and increased child mental health difficulties. Prior research supports higher father self-efficacy being associated with increased authoritative parenting [43] and increased authoritative parenting being associated with positive parent emotion regulation (e.g., parental acceptance) [45]. However, it was surprising to find greater parental acceptance of emotions to be associated with increased child mental health difficulties. The literature reveals perhaps this parental acceptance could be a passive, ‘detached approval’ of the child’s negative emotions without teaching the child how to regulate emotions themselves, leading to poorer child mental health symptoms [78,79,80] (p. 6). However, other scholars have found acceptance of a child’s negative emotions to be a positive parent regulation strategy [52] where such emotion-focused strategies lead to decreased child externalising and internalising symptoms [81]. This finding highlights the importance of ensuring emotion-focused parenting incorporates problem-solving skills to assist the child in developing the regulation of their own emotions. Future research could test if differences appear between passive parental acceptance and active parental acceptance involving problem-solving skills of children’s negative emotions to confirm or deny this hypothesis. Additionally, given that we found a potential suppression effect with the inclusion of the acceptance of emotions variable as a second mediator, further replication studies are required to clarify findings.

The authoritarian parenting–avoidance of emotions pathway was such that lower levels of father self-efficacy were associated with increased authoritarian parenting, which in turn was associated with increased avoidance of the child’s negative emotions and decreased child mental health difficulties. Prior studies in the literature support lower PSE being associated with controlling parenting [82], and harsher parenting being linked with poorer parent emotion regulation strategies (e.g., parental avoidance) [45]. However, it was surprising again to find that increased parental avoidance was linked with lower child mental health difficulties. The literature reveals parental avoidance may be linked with parents’ utilisation of maladaptive problem-solving strategies such as distraction and dismissing to erase the child’s negative emotions, which may be beneficial in some contexts but not all [52]. Participants in this study were a community sample of fathers and may have reported fewer extreme forms of authoritarian parenting and avoidance of emotions compared to other parent groups due to the desire to appear in line with what is considered by society to be good parenting. This could potentially describe why better child mental health outcomes were observed. Furthermore, since there were low mean scores on the authoritarian parenting scale from the community sample, those who scored higher on this scale may be more representative of mid-level rather than high-level authoritarian parenting. Yet, since our findings contrast with prior research revealing parents who dismiss their children’s negative emotions is linked with increased child depressive symptoms, future research is needed [48,83]. Future research could build on our significant findings and distinguish in what contexts fathers’ avoidance of children’s negative emotions becomes harmful to children.

Two other possible reasons may explain the surprising findings in the authoritative–acceptance and authoritarian–avoidance pathways. First, father-reported data on child mental health difficulties may be biased by the father’s perceptions and awareness of their child’s symptoms [72,83]. Australian research found approximately one-third (35%) of parents are confident in recognising child mental health symptoms [84]. This indicates that fathers who avoid their children’s negative emotions may be less sensitive to recognising and reporting child mental health symptoms and vice versa for father acceptance of child emotions. Secondly, fathers may have engineered a child-rearing environment that either encourages or suppresses the child’s emotional expression [85], potentially influencing child symptom visibility. Thus, future research should incorporate multi-dimensional tools (e.g., other-parent and teacher reports) for collecting data on child mental health symptoms.

### 4.1. Theoretical Implications

Our findings support the pathway of influence from fathers’ personal characteristics (father self-efficacy) to fathers’ parenting behaviours (parenting style and parent emotion regulation) and subsequently to child development as theorised by Cabrera, Fitzgerald, Bradley, and Roggman [6] and illustrated in Figure 1. We found three significant indirect pathways according to this model within the father–child dyad. This indicates that future research should test salient variables such as father self-efficacy with exterior familial variables (e.g., family relationships) included in Cabrera’s (2014) family–systems framework. Consistent with Pleck (2010), we found that decreased parental control (permissive parenting) was significantly associated with poorer child mental health outcomes and should be intervened in. However, we did not find support for positive parenting behaviours (e.g., authoritative parenting), contributing to fostering healthy developmental outcomes in children as theorised in both studies by Cabrera, Fitzgerald, Bradley, and Roggman [6] and Pleck [20]. Future research can build on this current study by demonstrating significant indirect positive parenting pathways between father self-efficacy and child mental health difficulties in the context of other salient familial variables involved in child wellbeing.

### 4.2. Practical Implications

This current study offers several practical implications for promoting positive child mental health development in communities. First, our findings of increased father self-efficacy being linked with decreased child mental health difficulties supports the promotion of parenting interventions for fathers such as the Triple P, Confident Parents, and Dads Tuning in to Kids programmes, as presented earlier [23,36,54]. Our findings suggest that these parenting interventions should target fathers low in parental self-efficacy, those high in permissive parenting, and those experiencing difficulties in parent emotion regulation. Since parent emotion-coaching programmes have been adapted for fathers (e.g., the Dads Tuning in to Kids programme) and have been found to benefit parent emotion regulation, child externalising symptoms, and fathers’ emotional awareness [54], we suggest that parenting style and PSE intervention programmes could adapt father-only courses as well. This may alleviate barriers to father engagement as qualitative research suggests fathers prefer father-only programme groups and may benefit from father-inclusive advertising [86]. This may also improve the effectiveness of parenting interventions for fathers, given that previous programmes have shown reduced effectiveness for fathers compared to mothers [8]. Finally, our findings support clinical mental health care providers promoting father self-efficacy as a tool that can yield downstream beneficial outcomes on children’s mental health [8].

### 4.3. Limitations and Future Directions

Although several limitations and suggestions for future directions have been mentioned, additional ones are discussed here. First, the ordering of variables in our parallel–sequential cross-sectional model was informed by prior research, but causal inferences cannot be drawn [30]. Our study did not test bi-directional influences, which have been theorised to exist in father–child dyads [6,20]. Given that PSE can be improved through positive child behaviours [72], future longitudinal research is required to test these links in the context of the other salient parenting variables discussed. Our cross-sectional model accounted for 18.5% of the variance in child mental health difficulties; thus, it will be important to incorporate further salient variables that influence parenting and child development into future models.

Second, our sample was mostly white, married, American fathers from a Western, well-educated background, limiting the generalisability of findings to populations of different cultures and family structures (e.g., single-parent families). Meta-analytic research suggests there are cross-cultural similarities in parenting because these parent behaviours are ‘universally adaptive’ for child development, but there are also differences due to environmental influences, cultural values, and norms [87] (p. 475). Therefore, there is a need for future cross-cultural fathering studies and research consisting of diverse samples of different family structures. Yet, due to the underrepresentation of fathers in parenting research as discussed earlier [8], our sample gave us the ability to report on father-specific results as opposed to mothers or parents in general, which has been the norm in the past [1,2]. Additionally, our sample of fathers reported on their school-age children (4–12 years old), where previous studies in the literature have focused on earlier child developmental periods (infants and toddlers) or later (adolescents) [17].

Third, our study did not control for the influence of mothers as co-parents on father self-efficacy and parenting behaviours. Earlier we discussed that mothers and fathers complement each other in overall child rearing [6,7]. Research posits individual parents as influential on the other parent’s behaviours and PSE beliefs [88,89]. This suggests maternal influence may be important to include or control for in future father self-efficacy studies. However, research also situates individual parents (mothers and fathers) as not significantly influencing the other parent’s beliefs and behaviours as much as themselves, providing further justification towards this current study [88].

Fourth, all data were self-reported by the fathers, potentially biasing the child’s mental health data, as stated earlier. Future research could conduct a replication study of this current study and utilise additional other parent, child, or teacher reports to facilitate a collection of child mental health difficulties. Additionally, future research could focus on the interactions between parenting beliefs, behaviours, and children’s mental health in the fourth parenting style of negligent parenting [38]. Furthermore, future research could attempt to capture data from those higher in the authoritarian parenting styles, as this current study was restricted in range for this particular parenting style. Utilising alternative methods (e.g., children’s retrospective reports, co-parent reports, or observed behaviours) may be useful in capturing the negligent parenting style and more extreme forms of authoritarian parenting.

Finally, as there were many significant inter-correlations found between the parenting styles and parent-facilitated emotion regulation strategies, only three significant indirect effect pathways were established. This suggests it may be beneficial for future research to consider a person-centred analysis rather than a variable-centred analysis to capture the grouping and clustering effects of parenting behaviours and dimensions.

## 5. Conclusions

In conclusion, this current study aimed to understand the relationship between father self-efficacy and child mental health difficulties and if this occurred through parenting style and parent emotion regulation. The findings highlighted father self-efficacy significantly influenced child mental health difficulties both directly and indirectly through permissive parenting, authoritative parenting–acceptance of a child’s negative emotions, and authoritarian parenting–avoidance of a child’s negative emotions. Yet to our surprise, the authoritative–acceptance pathway contributed to increased child mental health difficulties, and the authoritarian–avoidance pathway contributed to decreased child mental health difficulties. This demonstrates the need for future longitudinal research to understand the relationship between fathers’ parenting practices, emotion regulation strategies, and child mental health development to build on this study’s findings.

## Figures and Tables

**Figure 1 ijerph-22-00011-f001:**
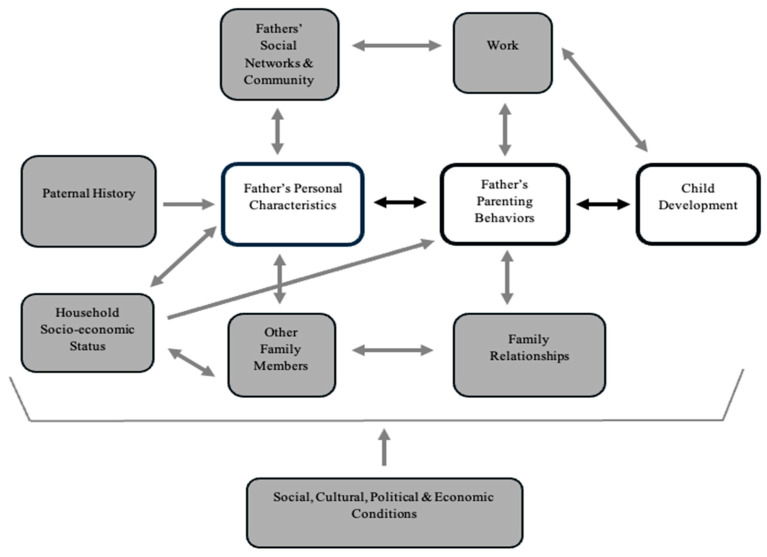
Adapted expanded model of paternal influences on child development. Note: The figure displays direct and indirect pathways by which fathers can influence their child’s development, such as the indirect pathway of interest highlighted in white. Adapted from Cabrera, Fitzgerald, Bradley, and Roggman [6]. Copyright 2014 by the National Council on Family Relations.

**Figure 2 ijerph-22-00011-f002:**
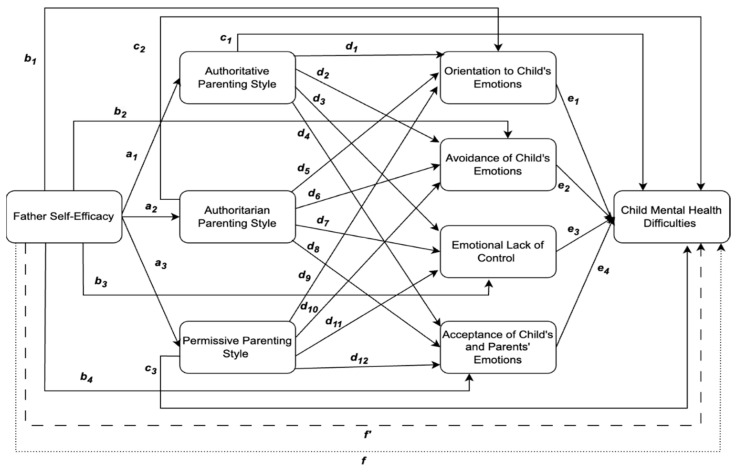
Proposed mixed parallel–sequential model of the relationship between father self-efficacy and child mental health difficulties. Note: The regression coefficients of the different paths in the conceptual model are represented by *a*_1_, *b*_1_, *c*_1_, and so forth. The relationship between father self- efficacy and child mental health difficulties is hypothesised to operate through a direct pathway, shown by the dashed line (*f*′), and indirect pathways. The indirect pathways are hypothesised to occur through one indirect effect variable—either a parenting style or parent-facilitated emotion regulation variable—or sequentially through two of these indirect effect variables. The total effect of father self-efficacy on child mental health difficulties is shown by the dotted line (*f*).

**Figure 3 ijerph-22-00011-f003:**
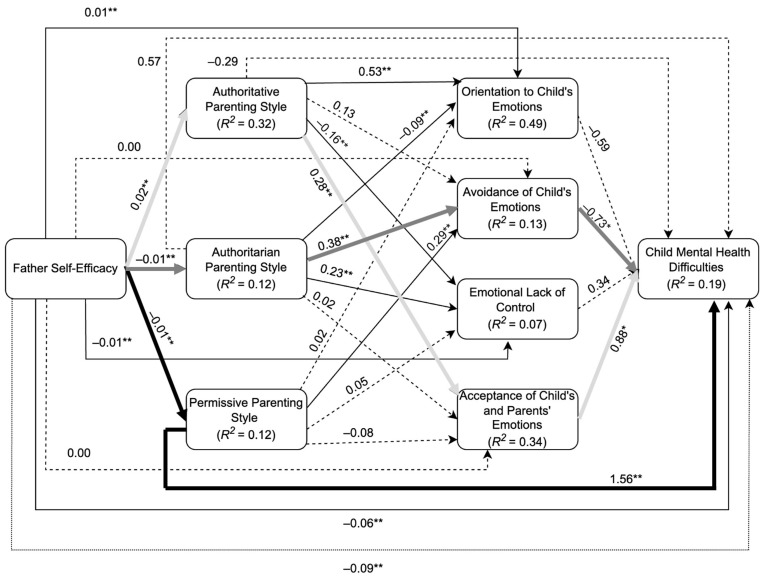
Path analysis model of associations between father self-efficacy and child mental health difficulties via parenting style and parent emotion regulation strategies. Note: Path diagram of the parallel–sequential model after accounting for control variables. Standardised coefficients are presented along arrows between variables. Continuous arrows with values ** *p* < 0.001, * *p* < 0.05 imply significant pathways, while dashed lines imply non-significance. Three significant partial indirect pathways are shown by bolded arrows, permissive parenting (black bolded arrows), an authoritative–acceptance pathway (light-grey bolded arrows), and an authoritarian–avoidance pathway (dark-grey bolded arrows). The total effect (−0.09 **) is indicated through the dotted line. (*N* = 350).

**Table 1 ijerph-22-00011-t001:** Descriptive statistics for primary and control variables.

Variable	M	SD	Min	Max	Skewness	Kurtosis	α
1. FSE	157.77	21.16	67.05	189.41	−0.83	0.79	0.93
2. Child Mental Health Difficulties	7.85	5.55	0.00	26.00	0.55	0.13	0.88
3. Authoritative	3.95	0.61	2.13	5.00	0.13	−0.26	0.88
4. Permissive	2.13	0.68	1.00	4.40	0.67	0.23	0.66
5. Authoritarian	1.78	0.63	1.00	4.50	1.53	3.08	0.86
6. OCE	3.18	0.66	0.80	4.00	−0.79	0.55	0.85
7. ACE	1.78	0.93	0.00	4.00	0.25	−0.81	0.84
8. ELC	1.07	0.69	0.00	2.60	0.29	−0.77	0.65
9. ACPE	2.43	0.75	0.75	4.00	−0.02	−0.59	0.61
10. Child’s Age	7.50	1.99	4.00	12.58	0.54	−0.80	-
11. Child’s Gender	-	-	-	-	-	-	-
12. Number of Children	3.30	1.03	2.00	9.00	1.30	3.53	-

Note. FSE = father self-efficacy; OCE = orientation to child’s emotions; ACE = avoidance of child’s emotions; ELC = emotional lack of control; ACPE = acceptance of parent’s and child’s emotions. PSDQ subscales of authoritative, permissive, and authoritarian were measured on a scale from 1 to 5. PERS subscales of OCE, ACE, ELC, and ACPE were measured on a scale from 0 to 4. Child’s sex was scored as 0 = male, 1 = female. Child’s age, child’s sex, and number of children acted as control variables. *N* = 350.

**Table 2 ijerph-22-00011-t002:** Correlation matrix for primary and control variables.

Variable	1	2	3	4	5	6	7	8	9	10	11
1. FSE	-										
2. Child Mental Health Difficulties	−0.35 **	-									
3. Authoritative	0.56 *	−0.23 **	-								
4. Permissive	−0.32 **	0.27 **	−0.15 **	-							
5.Authoritarian	−0.32 **	0.22 **	−0.23 **	0.50 **	-						
6. OCE	0.56 **	−0.27 **	0.65 **	−0.18 **	−0.27 **	-					
7. ACE	−0.04	−0.04	0.04	0.31 **	0.33 **	0.14 **	-				
8. ELC	−0.51 **	0.24 **	−0.38 **	0.30 **	0.38 **	−0.36 **	0.18 **	-			
9. ACPE	0.17 **	0.04	0.25 **	−0.10	−0.08	0.19 **	−0.08	−0.05	-		
10. Child’s Age	0.07	0.00	0.07	−0.88	−0.01	0.06	0.03	−0.12 *	−0.04	-	
11. Child’s Gender	0.02	−0.00	0.07	−0.03	−0.11 *	0.04	−0.08	−0.03	0.02	−0.02	-
12. Number of Children	0.02	−0.04	−0.06	−0.09	−0.02	−0.05	−0.12 *	−0.12 *	−0.02	0.06	−0.03

Note. Unstandardised correlations between all variables included in the model. Bold values with ** *p* < 0.001, * *p* < 0.05 implies statistical significance. FSE = father self-efficacy; OCE = orientation to child’s emotions; ACE = avoidance of child’s emotions; ELC = emotional lack of control; ACPE = acceptance of parent’s and child’s emotions. Child’s sex was scored as 0 = male, 1 = female and, along with child’s age and number of children, acted as control variables. *N* = 350.

## Data Availability

A summary of the data presented in this study is available on request from the corresponding authors. The sharing of disaggregated, individual-level data is not permissible as a requirement of the governing Human Research Ethics Committee.

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
