# Peer review of "The Indirect Effects of Fathers’ Parenting Style and Parent Emotion Regulation on the Relationship Between Father Self-Efficacy and Children’s Mental Health Difficulties"

_ijerph, 2024, doi:10.3390/ijerph22010011_

Round 1

Reviewer 1 Report

Comments and Suggestions for Authors

See attached document for comments

Author Response

Thank you for the helpful feedback on this study. Please see the attachment for a full list of how each comment was considered and addressed. 

Reviewer 2 Report

Comments and Suggestions for Authors

I suggest to author, using newest references the latest 3 year ( 2021 to 2024).

Author Response

(The authors gave the same response as above.)

Reviewer 3 Report

Comments and Suggestions for Authors

The topic described by the authors is very interesting and topical, especially at a time when the subject of fatherhood is becoming more and more prevalent and raised in scientific, political and economic discussions. The figure of father is visible and appreciated more and more in the educational process. There is also more and more research on fatherhood and its impact on the parenting, upbringing, family process. Changes to the law, increasing the possibility of paid leave for fathers, extending leave for them, organisations for fathers and those fighting for fathers' rights are confirmation of positive changes in the area. Thus, the topic of the article and the research undertaken are extremely important.

The methodology proposed by authors is justified. The tools used allow us to obtain results that respond to the hypotheses set. The hypotheses are set correctly, although the proposed model (Figure 2) is quite complex to analyse freely.

The description of the results as well as the other parts of the article are described in an interesting way. However, I suggest supplementing the manuscript with a few points: 

1. If possible, I suggest simplifying the hypotheses of the conceptual model in Figure 2. 

2. Although the description of the study includes the different number of children the subject has, the number of children does not appear as an indirect effect (section 2.1), so this information should be supplemented.

3. As the authors point out, there are four parenting styles (authoritative, authoritarian, permissive, negligent parenting style). To be honest it’s hard to agree that the negligent parenting style has been described in the literature for a short time. Even the authors' reference to the literature suggests 1983, hence it is an oversight in the study not to include the fourth parenting style. It is therefore necessary to justify in more detail the incomplete range of parenting styles investigated (section 1.4) and to propose it in the Limitations and Future Directions (section 4.3) a study of the full range of parenting styles, which would make the results of the study more interesting and complete in the context of developing support programmes for parents/fathers. It can be assumed that the neglectful style may have the greatest impact on the child's mental health, hence its inclusion is important from the point of view of putting the data obtained into practice.

4. It is worth rethinking a topic that seems to be too long and complicated. I propose the topic of the article: Indirect Effects of Fathers’ Parenting Style in the context of/and children's mental health difficulties. However, changing the topic is a free decision of the authors.

With minor changes, I recommend that the manuscript can be published. 

Author Response

(The authors gave the same response as above.)
